# What Can 5G Do for Public Safety? Structural Health Monitoring and Earthquake Early Warning Scenarios

**DOI:** 10.3390/s22083020

**Published:** 2022-04-14

**Authors:** Fabio Franchi, Andrea Marotta, Claudia Rinaldi, Fabio Graziosi, Luciano Fratocchi, Massimo Parisse

**Affiliations:** 1Department of Information Engineering, Computer Science and Mathematics, Università degli Studi dell’Aquila, 67100 L’Aquila, Italy; fabio.franchi@univaq.it (F.F.); andrea.marotta@univaq.it (A.M.); fabio.graziosi@univaq.it (F.G.); 2Research Unit L’Aquila, National Inter-University Consortium for Telecommunications (CNIT), 67100 L’Aquila, Italy; 3Department of Industrial and Information Engineering and Economics, Università degli Studi dell’Aquila, 67100 L’Aquila, Italy; luciano.fratocchi@univaq.it (L.F.); massimo.parisse@univaq.it (M.P.)

**Keywords:** 5G, structural health monitoring, early warning, IoT, multi-access edge computing, network slicing

## Abstract

The 5th generation of mobile networks has come to the market bringing the promise of disruptive performances as low latency, availability and reliability, imposing the development of the so-called “killer applications”. This contribution presents a 5G use case in the context of Structural Health Monitoring which guarantees an unprecedented level of reliability when exploited for public safety purposes as Earthquake Early Warning. The interest on this topic is at first justified through a deep market analysis, and subsequently declined in terms of public safety benefits. A specific sensor board, guaranteeing real-time processing and 5G connectivity, is presented as the foundation on which the architecture of the network is designed and developed. Advantages of 5G-enabled urban safety are then discussed and proven in the experimentation results, showing that the proposed architecture guarantees lower latency delays and overcome the impairments of cloud solutions especially in terms of delays variability.

## 1. Introduction

The fifth generation (5G) of the mobile access technology is changing the mobile communication business ecosystem by introducing specific services that can be operated by different stakeholders in various fields, i.e., public safety, healthcare, energy, manufacturing, media and entertainment, financial services, public transport, retail, automotive, and agriculture.

This contribution focuses on the public safety vertical. These kinds of applications are referred to as Public Protection and Disaster Relief (PPDR) and, based on the 5G Observatory, only five trials, out of 181, were dedicated to PPDR [1].

These applications require communications efficiency, especially under harsh conditions when the safety of critical infrastructures may not be guaranteed due to catastrophic events. This pushes the investigation toward solutions that are able to guarantee all the services to properly address the crisis.

Introducing 5G in this context means overcoming the limited flexibility and scalability of legacy systems that are mainly due to hardware-based networks functions exploitation and the absence of a proper orchestrator [2]. This is possible by exploiting emergent technologies within the 5G paradigm as (i) network slicing, which may guarantee dedicated services for applications with different requirements; (ii) Network Function Virtualization (NFV) minimizing the dependence on hardware constraints, thus allowing exploitation of constantly update and reliable devices for public safety; (iii) Software Defined Network (SDN) enabling proper management of network resources, which is particularly useful when critical conditions introduce communication network failures; (iv) Multi-access Edge Computing (MEC), which drastically reduces the latency and backhaul/core network traffic, thus improving the reliability of the emergency communication/response; (v) 5G Radio Access Network (RAN) to enable unprecedented flexibility and performance guarantees tailored to specific services.

Moreover, the shared infrastructure approach and the use of public instead of dedicated networks, both offered by the 5G scenario, have the potential to overcome the problem of affordability and economy of scale, which has caused the public safety sector to remain many technological generations behind the public communication solutions.

Various trials to exploit the previously discussed potentials have been designed and developed in recent years. The 5GENESIS (ICT-17-2018) platform was tested in Malaga by furnishing the local police commercial mobile phones for live videos exchange to receive and transmit live video in the coverage area of the deployed 5G network. A comprehensive document on experimental results is not yet available, but the experimentation methodology is described in [3]. A solution exploiting 5G technology for air quality monitoring services for smart cities has been developed in 5G EVE project (ICT-17-2018) [4]. In this case, 5G unprecedented reliability preserves public safety by allowing immediate reaction in case of air quality degradation. A trial for guaranteeing increased quality of service for military purposes has been conducted within the project 5G-VINNI (ICT-17-2018), where the paradigm of Network Slicing has proven to be fundamental for virtually separating Military and Commercial Networks to reserve high reliability for military communications [5]. Network slicing for improving public safety is also the main purpose of the project METRO-HAUL (ICT-7-2016), where the full workflow from planning, to orchestration, deployment, and running a network slice over an edge-computing enabled metro optical networking test-bed has been successfully carried out [6]. The concept of prioritizing uplink data traffic for public safety services has been successfully proven within the test cases proposed in 5G-VICTORI (ICT-17-2018) [7]. Within the project 5G-XCAST (ICT-17-2018), a 5G based solution for public warning through multimedia has been tested, successfully managing a severe amount of data in a short elapsed time [8]. In the trial that has taken place within the 5G ESSENCE (ICT-7-2016) project in Newcastle, the potentialities of 5G managed to assign communications priorities to first responders in a crowded scenario [9]. Another test case covering the vertical of emergencies requiring real-time communications between first responders (i.e., ambulances and doctors) is the scope of the 5G TRANSFORMER project (ICT-2016-2) [10].

Many other projects concerning with PPDR solutions though 5G technologies could be cited as successful examples, but based on the authors’ understanding, the solution integrating two widely different applications that is experimented here, and whose architecture was proposed in [11], is not yet exploited in the current literature.

In this paper, the first experimental outcomes of an application related to a 5G-enabled IoT for public safety is presented. Generally, 5G-enabled IoT consists of various types of IoT devices sending data through a very efficient communication infrastructure guaranteeing low latency while conceiving with public safety or generally speaking, safety applications. The described solution for Structural Health Monitoring (SHM) of buildings and infrastructures addresses both normal and emergency conditions, and it is one of the outcomes of the experimentation within the 5G trial in the city of L’Aquila (Italy) [12]. Specifically, the presented scenario demonstrates how the 5G network supporting SHM becomes an enabling technology for PPDR applications and in particular for EEW. Based on the classification proposed in [1] of the 5G PPDR experimentations that are categorized depending on the offered services and on application use case families, the presented trial can be classified into the PPDR service category of *sensor services* and within the use case category of *higher reliability*, *availability*, and *lower latency*.

The remainder of the paper is organized as follows: in Section 2, the importance of SHM for maintenance and damages protections is discussed, while its economical impact is tackled in the market analysis presented in Section 3, followed by the opportunity offered by a SHM solution for public safety, presented in Section 4. The developed sensor board allowing the SHM integration with an early warning is briefly described in Section 5, while details on the enabling network architecture are given in Section 6. Finally, Section 7 discusses about lesson learned and experimentation results, and conclusions are drawn in Section 8.

## 2. Structural Health Monitoring

Based on [13], the SHM process can be defined in terms of four steps:operation evaluation,data acquisition, normalization and cleansing,feature selection and information condensation,statistical model development for feature discrimination.

How these four steps are implemented clearly depends on the specific application (e.g., seismic analysis, cracks, tilt growth, dynamic analysis), imposing specific constraints. For instance, measurements’ synchronization is required while conceiving with dynamic analysis oriented monitoring [14], while this condition can be relaxed when dealing with slowly varying monitored parameters.

Since SHM systems deal with safety of buildings and people, reliability represents a key requirement. On the other hand, the massive deployment of sensors stresses energy efficiency to guarantee sufficient lifetime to battery powered devices. Energy efficiency can be achieved by the design of low power devices and efficient power management strategies. Buildings are monitored by exploiting data coming from the deployed sensors in order to evaluate their performance over time, for instance in terms of safety, thus allowing the design of a proper restoration plan, to reduce their vulnerability. When a catastrophic event arises, constantly tracked structures are able to activate proper mechanisms for emergency response.

Traditional systems are based on wired grids of sensors deployed along a structure and have a high cost, considerable size and poor flexibility.

In recent years, a significant opportunity in the field of SHM technologies has been represented by the gradual growth of Wireless Sensor Networks (WSNs). Indeed, they offer the opportunity of reducing deployment costs and improving service flexibility by eliminating connection cables. Moreover, the reduction or even the total absence of cabled connections allows WSNs to be suitable for installation in buildings of historical or artistic relevance where wireless nodes can be properly hidden.

Furthermore, WSN nodes can be configured to analyze sensors measurements and eventually trigger alarm signals in case of damage detected.

For example, in the use case discussed in Section 4, each building involved in the test has been equipped with a network of seismic monitoring sensors. The aim of this application is to build a distributed monitoring network at the lowest possible expenses, which guarantees the opportunity of properly collecting, representing and analyzing the available data. A central collector receives sensors gathered data through a 5G network and a dynamic service creation approach is adopted for presenting these data to applications.

## 3. SHM Market Survey

One of the advantages of the scenario proposed in this paper is given by the exploitation of a SHM system, commonly intended for long-term purposes with relaxed constraints, for public safety. In this section, the importance of SHM from an economic point of view is demonstrated, which gives added value to the feasibility of the proposed scenario.

On a global level, the size of the market referable to SHM in 2020 is US$1.5 billion, and it is in a growing phase: the market is expected to reach a value of US$2.9 million in 2025 with a compound annual growth rate of 14.1 percent in this period [15]. At present, the market share consists of applications coming mainly from some sectors (i.e., aerospace, civil structures, oil & gas geophysics) with market data, highlighting the fact that the main SHM market is represented by applications for civil structures, which, with approximately US$2.586 million, will represent about 65 percent of the entire market in 2023.

A further segmentation element is represented by the type of connection between sensors, wired or wireless. From this point of view, the research highlights how the market is moving towards a wireless approach, to overcome the limits of wired technologies (i.e., lack of interconnection between sensors, high costs and installation times on existing structures), as shown by the growth rate about 5 percent in favor of wireless technology.

A last analysis is related to the registered patents for SHM with the aim of highlighting the trends and the distribution of patented technologies.

Results shown in Figure 1 have raised some evidence of interest:*measurement* mainly concerns the concepts of “deformation” and “vibration”;there is considerable interest in solutions related to the use of optical fiber;in terms of applications, the most represented categories are those related to the aeronautical sector, civil engineering and bridge monitoring;the category related to “measured data” highlights a multiplicity of variables considered in the patented solutions for the purposes of structural monitoring.

Moreover, the analysis found that the total number of patents worldwide (3035) is not particularly high and the 42 percent of world patents are held by 10 players.

## 4. SHM Supporting Urban Safety System

The development of 5G networks is of particular interest to the broad issue of mission critical services and emergency management systems. Traditionally, public safety communications services have been implemented through Personal Mobile Radio (PMR) systems. However, media transmission can be useful in many critical scenarios, but current PMR systems cannot support these types of services [16]. Therefore, there is a need for systems to support public safety communications characterized by higher transmission speeds. The evolution that has taken place in recent years in communications systems has given new impetus to the rethinking of such systems both in the public sector as well as through the experimentation of new business models. Indeed, the support of public safety applications is among the objectives of 3GPP Rel.17 [17]. The reasons for the failure in the evolution of traditional public safety networks are mainly due to the need for the use of dedicated infrastructures characterized by high costs, which prevented the densification and the enhancement of such systems. On this hand, 5G appears as a perfect candidate to overcome such limitations, as it allows for supporting a multitude of services with heterogeneous requirements over a common physical infrastructure, thus reducing required investments per service and maintaining isolation among the different services.

With particular focus on SHM, a challenging application is the utilization of the sensors not only for a monitoring purpose, but also for the early detection and notification of seismic events. This application, known as Earthquake Early Warning (EEW), has been studied and tested with the limitations of the dedicated communications systems described above [11]. However, the possibility to leverage an uRLLC (ultra-Reliable Low Latency Communications) infrastructure unveils tremendous potential benefits on safety of human beings and infrastructures. Figure 2 represents the proposed scenario for EEW, where the building sensing at first the earthquake immediately starts an information procedure to warn the surrounding structures of the incoming most critical phase. Once the alert is received, a series of action can be taken to prevent severe damages to structures and people such as initiation of elevator recall procedures to ground floor, exit doors unlocking, place sensitive equipment in safe mode, secure hazardous materials, halt production lines to reduce damage, switch on emergency lights, etc., as well as alerting the people into buildings. Trivially, the lower is the time required for the propagation of the warning, the higher is the effectiveness of such a system.

Figure 3 shows the travel time curves for the earthquake wave ($t_{e}$) and the early warning message ($t_{w}$). A *risk region* is defined as the area where the seismic wave arrives before the early warning, i.e., $t_{e}<t_{w}$ [18]. The adoption of the 5G network as supporting infrastructure can significantly reduce the area of the *risk region* by offering higher throughput and low latency and by leveraging the density of connected devices in contrast to legacy EEW systems with sensing stations placed at a distance of 10–30 km. It is worth mentioning that, due to the time required to collect the minimum data to detect the seismic event, there is a not compressible blind area, highlighted in *dark grey* in Figure 3.

## 5. Sensor Board Design and Data Processing

The exploitation of a system able to support both SHM and EEW scenarios relies on the setup of monitoring sensor networks at a building level able to collect data that characterize the structural behavior of the buildings. The same data will be used to detect a severe seismic event in order to generate an EEW-message.

An ad-hoc sensor board has been developed by authors in order to demonstrate the capabilities of such kind of systems designed as a high performance device for real-time SHM of buildings and infrastructures [19]. It is based on an ultra-low-power micro-controller, which offers numerous communications and high-performance interfaces and using MEMS accelerometers acting as a measurements unit for SHM and as a sentinel in the event of accelerations detected above a certain threshold. It can provide different network interfaces, wired and wireless, such as Ethernet (even for power supply through PoE) and W-MBUS protocol at 169 MHz or 868 MHz in order to communicate with other nodes of the SHM network, and it can be equipped with a 5G expansion module.

The board is equipped with a 4-channel 24-bit ADC able of sampling at 100 Hz. However, on-board data processing capabilities allow for minimizing the total amount of data to be transmitted. It also provides the possibility to handle measurements coming from other kinds of sensors like tilt-meters and crack-meters. The integrated temperature sensor allows for evaluating the thermal effect on the structure and on the sensor, thus allowing for distinguishing thermally induced variations from real measurements.

Data acquired from the sensors feed a data-driven model, based on System Identification from Control Theory and Machine Learning (ML) from Computer Science, [20]. Using such data, we were able to derive, with the algorithm developed in [21] and using only regression trees, dynamical models with a high accuracy of the vibrations induced on the structure by an earthquake reproduced via mechanical actuators. Our identified models have been used to construct an optimal predictive control algorithm (Model Predictive Control—MPC) in order to reduce the oscillations in terms of accelerations of the structure by means of active dampers. Subsequently, these results have been extended [22] also using Random Forests and Neural Networks, showing that the use of Random Forests allows for further reducing the acceleration of the structure compared to the results obtained using only Regression Trees, considerably reducing the amount of kinetic energy involved in the process and in particular the effort required by the dampers. The developed model identification techniques have also been exploited in the context of damage detection. Current works are based on the identification algorithms of the models developed were applied by comparing them with the Principal Component Analysis techniques, appropriately combined with Kalman filtering theory. In addition, sensor selection algorithms have been developed based on the concept of Entropy and Information Gain, showing how in some cases there is the possibility to reduce the number of sensors significantly, while improving at the same time the accuracy of the predictive model.

## 6. 5G, Network Slicing and MEC to Support SHM and EEW

An enabling paradigm to meet performance and security requirements of SHM and EEW is represented by network slicing that is a tool for the implementation of vertical and transparent support of services with different requirements (i.e., slices) in terms of throughput, latency, and reliability through a shared physical infrastructure.

Referring to the architecture shown in Figure 2, three network segments can be identified which contribute to the experienced latency and reliability for the services under consideration, namely the *mobile segment*, the *transport segment* and the *core segment*. The definition of the slices hosting respectively the SHM and EEW services involves the configuration of those three segments to adequately handle the traffic belonging to the slices. In particular, the mobile segment has to efficiently manage radio resources. While the monitoring traffic of SHM can be handled with a best effort approach, the EEW traffic is characterized by more stringent requirements.

Due to the uRLLC nature of the EEW slice, the mobile network can adopt a series of strategies to reduce the experienced latency of early warning messages such as: grant free transmission, slot duration reduction, semi persistent scheduling, and uRLLC-eMBB multiplexing. Transport segment has to forward the traffic pertaining to the slices through the radio access network segments (i.e., fronthaul, midhaul and backhaul), making use of the packet and optical network infrastructures.

With respect to previous mobile communications technologies, 5G allows for offering specific reliability levels per service. This can be achieved by leveraging multi-connectivity, i.e., the possibility to connect to multiple radio access points, ad-hoc transmission strategies [23], and by adopting protection strategies at the optical transport segment as studied in [24].

Referring to the core segment, computation elements also referred to as Virtual Network Functions (VNFs) have to be orchestrated according to service metrics.

The configuration of network and computing resources for different slices is performed through the use of SDN controller for the network configuration and MANagament and Orchestration (MANO) for the reservation of computing resources and the deployment of the VNFs.

On this hand, MEC represents a useful tool to redirect the traffic belonging to a specific service towards the edge of the network and closer to the terminals by enabling different types of applications such as novel multimedia services [25], enhanced vehicular communications [26], and industrial automation [27]. Instead of sending whole data to the remote cloud for processing, the edge cloud analyzes, processes, and stores it. With respect to the EEW and SHM services, traffic belonging to the EEW is elaborated at the edge while the SHM traffic is processed at a remote cloud. A graphical overview is given in Figure 4. The gathered data of each monitored building are sent periodically to a remote infrastructure (remote cloud) which takes care of storing data over the long term, allowing the subsequent data processing by complex algorithms. Once a critical event is detected, the traffic is directed through the EEW slice to a local infrastructure (edge cloud), which, despite having limited capacity, is able to identify critical events based on the data received and promptly launch the necessary alarms, guaranteeing proper availability and reliability levels.

Differently from the previous mobile generation 4G, 5G natively supports service deployment at the edge thanks to the introduction of the novel User Plane Function (UPF), which represents an anchor element between the mobile and data network. As shown in Figure 4, by deploying an instance of the UPF at the edge, the user plane traffic can leave the mobile network after reaching the 5G base station and be redirected towards the Edge Cloud. Note that the exploitation of MEC in 4G mobile networks is only possible with the deployment of private mobile networks with dedicated instances of Evolved Packet Core. Private networks, however, represents an unfeasible solution for geographically distributed public safety services such as SHM and EEW for radio planning and cost reasons.

The considered scenario adopts the Message Queue Telemetry Transport (MQTT) protocol at the application layer, which is based on the publish/subscribe pattern. Based on this pattern, buildings act as publishers and send early warning messages or monitoring data to a so-called “broker”. The broker sends collected data to the subscribers, which are represented by the buildings interested by the critical event. A graphical representation of this process is given in Figure 4.

## 7. Lesson Learned with the Italian 5G Trial: Experimentation and Results

The test-bed adopted for the 5G use case that took place in the city of L’Aquila, addressing the SHM and EEW scenario, implements the architecture shown in Figure 4. The radio infrastructure was based on ZTE Radio and Distributed Units operating with a 100 MHz system bandwidth in the frequency interval assigned to the operator Wind3 for 5G trial purpose, i.e., 3.7–3.8 GHz.

The edge cloud was located within the city of L’Aquila, and it was reachable by the gNodeB through the metro network. The remote cloud was located in proximity of the city of Rome (i.e., almost 100 km far away from gNodeB), and it was reachable through the operator’s national backbone network.

As shown in Figure 2 and Figure 4, each sensor board, described in Section 5, was connected to an 5G CPE through a wired Ethernet link while a suitable embeddable 5G modem was not available to the market during the above-mentioned 5G trial.

Tests have been conducted using a steel test frame, shown in Figure 5, representing a concrete building with a regular plan equipped with four sensor boards enabling both SHM and EEW scenarios.

A commercially available traffic generator was utilized to perform network latency measurements and characterize the delay introduced by the network infrastructure for the metro network connecting the base station to the edge cloud and operator’s national backbone utilized to reach the Remote Cloud. Measured network latencies at the metro network (in ms) follow a Gaussian distribution $D_{\mathrm{edge}}∼N\left(2.1,\;0.9\right)$, while the ones of the geographical network also follow a normal distribution $D_{\mathrm{cloud}}∼N\left(19.75,\;3.1\right)$.

Sensors data were subsequently sent to the edge and remote cloud and delays were assessed. Publish messages were generated every 10 ms by the publisher, with MQTT QoS set to zero. Figure 6 shows the time at which data are available for publishing and the reception times at the edge cloud (i.e., EEW alarm) and at the remote cloud (i.e., SHM monitoring). Concerning the edge cloud, data are received with a reduced delay, due to the lower metro network delay. Results demonstrate that latency has to be taken into account during the system design phase, in order to properly cope with the variability it introduces.

An ad-hoc client software was developed to emulate the behaviour of the sensor board and measure latency at the application layer. To measure the latency at the application layer, a time-stamp-based mechanism was introduced in the exchange of messages between application components, and GPS was utilized to synchronize the machines involved in the measurements. Measured delays at the application layer approximately follow a Gaussian distribution with $D_{\mathrm{edge}}∼N\left(4.2,\;1.3\right)$, $D_{\mathrm{cloud}}∼N\left(40,\;8\right),$ whose representation is given in Figure 7. Trivially, this means that the remote scenario pays a higher penalty considering the aggregate delay (i.e., network plus application delay) with respect to the edge one. The higher reliability of the edge solution is demonstrated by the very low variance of the delay distribution, showing that avoiding the core network allows for overcoming its impairments especially in terms of delays variability. It is worth mentioning that the results related to the Remote Cloud solution are comparable with a standard deployment based on 4G mobile connectivity, which requires the traffic to reach the core network, and thus increases end-to-end latency. Results show that the main advantage deriving from the adoption of a 5G architecture for the SHM stands in the native support of MEC deployment which allows for drastically reducing latency and reliability for the EEW service.

In terms of seismic waves and EEW messages propagation times, it could be difficult to evaluate the impact of the reduction of the elapsed time to propagate the information wave related to the EEW message because it depends on many aspects characterizing the seismic wave propagation such as geological conditions, constructions density, population density and many others. However, considering that the average speed of a superficial seismic wave is about 8 Km/s (referred, for example, to the Southern Tyrrhenian subduction zone [28]), the deployment of a 5G-enabled edge solution for EEW is able to increase the alerted area of hundred meters compared to the remote cloud one for a single source of the alert message.

The geographical distribution of the buildings may involve multiple edge computing nodes. Figure 8 shows the impact of density of edge nodes on the experienced latency at the application layer, in a scenario where adjacent MEC resources are utilized as backups in case of failure. Results highlight that increasing the density reduces the latency and the difference between working and backup MEC resources. Furthermore, the variance of delay decreases with higher density, making the MEC solution more reliable. Even if such density of MEC infrastructures comes at a higher cost, it is worth mentioning that such costs are shared among all the services deployed at the edge.

Redundancy of MEC infrastructures represents an open research challenge and different strategies can be implemented to reduce, for example, the need for coordination (and delay) among edge nodes, considering different metrics, e.g., cost, risk, bandwidth usage, etc., [26].

Moreover, in terms of availability and reliability, the remote cloud scenario is exposed to severe service degradation or outages, related or not to the seismic event, due to its design principle (i.e., best-effort) not compliant with PPDR services requirements. On the contrary, an MEC solution is able to bypass unpredictable drawbacks that would indeed introduce unacceptable latencies providing to network operators a viable way to increase PPDR Service Level Agreement (SLA) compliancy.

Finally, an MEC approach allows an end-to-end control of the traffic of the EEW service and enables the network operator to apply traffic-specific policies to ensure guaranteed performance (in contrast with the remote cloud approach). As a lesson learned, critical applications with a public safety scope as the EEW also have to deal with multi-operator strategies to make elaboration at the edge available for users belonging to a multiplicity of operators.

## 8. Conclusions

This contribution discussed the potential role of 5G for public safety applications, presenting a scenario allowing the implementation of both SHM and EEW solutions based on a common sensor device and a shared communication infrastructure.

The concept of SHM is at first decreased in terms of definition and future directions and subsequently a market analysis is presented to demonstrate the market demand for buildings’ monitoring solutions. The scenario involving SHM for public safety is then presented and its challenges discussed. The latter issues are partially solved by a specific implementation of a sensor board allowing the variety of sensors and connectivity technologies that may be needed in this context. Then, novel architectural solutions able to respond to the discussed challenges are presented. Results demonstrate that public safety issues, especially when specified in terms of EEW, may gather significant advantages in terms of latency reduction and reliability, when novel technologies, such as 5G, MEC, and network slicing, are used.

As a matter of future works, the potential of the presented scenario should be analysed by focusing on the opportunity offered by the sixth generation (6G), together with IoT and ML for implementing a truly smart city [29].

## Figures and Tables

**Figure 1 sensors-22-03020-f001:**
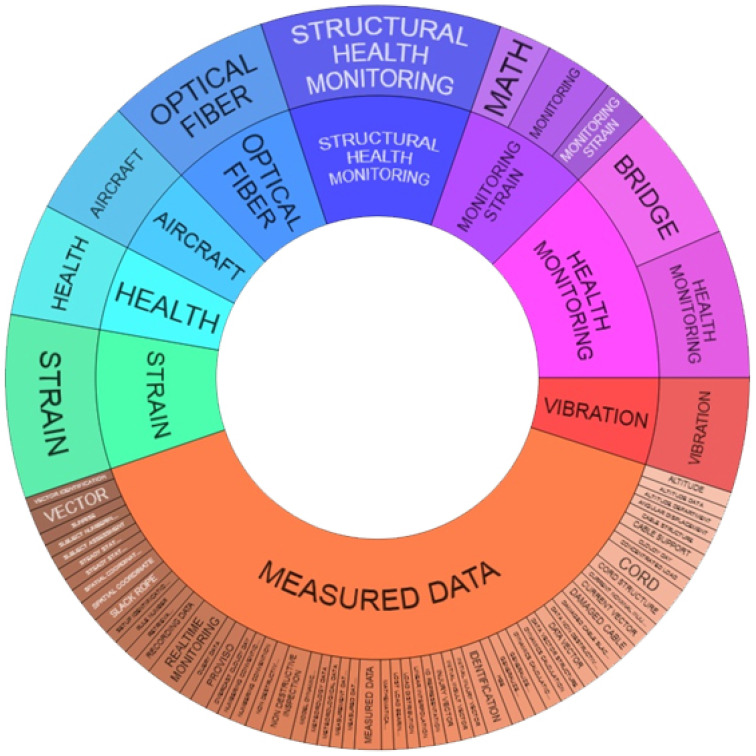
Patent data elaboration, University of L’Aquila, 2020.

**Figure 2 sensors-22-03020-f002:**
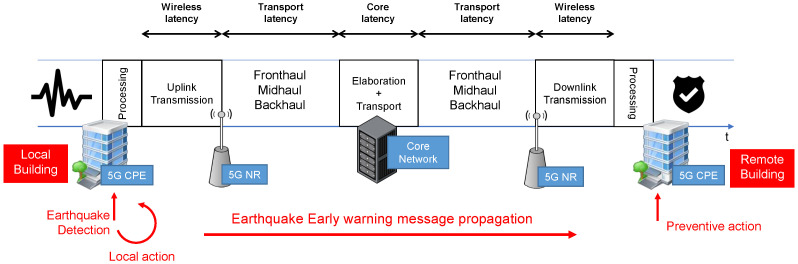
Reference architecture for the 5G-based EEW system.

**Figure 3 sensors-22-03020-f003:**
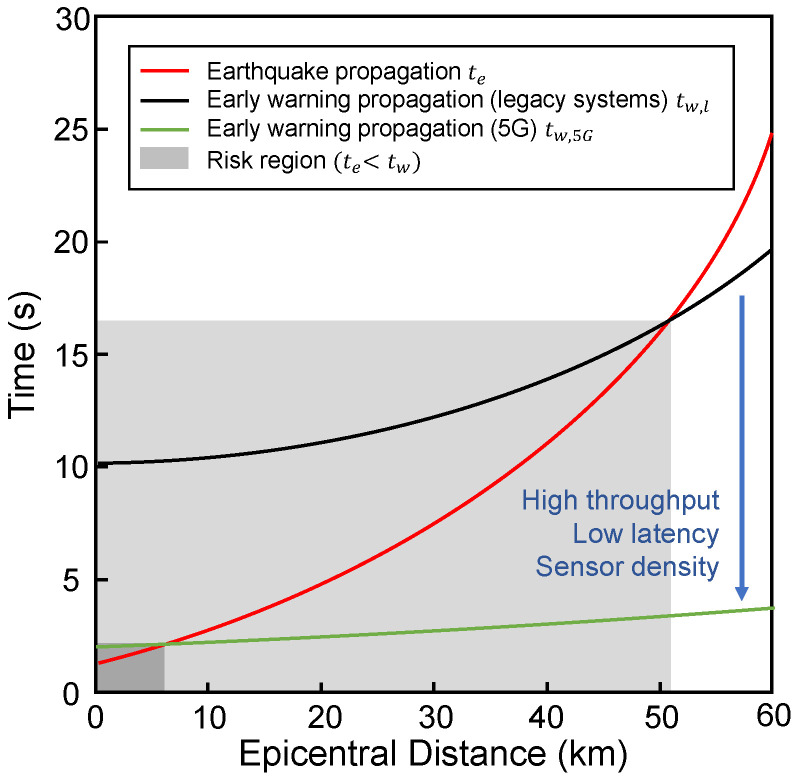
Seismic wave travel time vs. EEW message travel time. Adapted from Tajima et al. [18].

**Figure 4 sensors-22-03020-f004:**
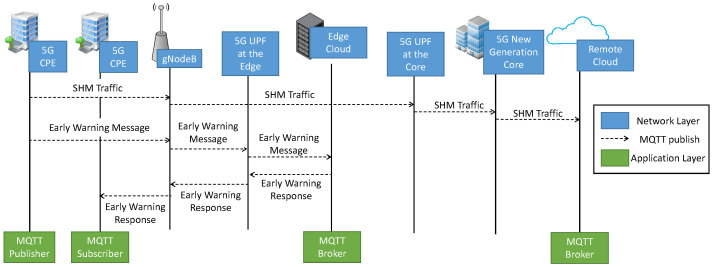
MEC for SHM and EEW.

**Figure 5 sensors-22-03020-f005:**
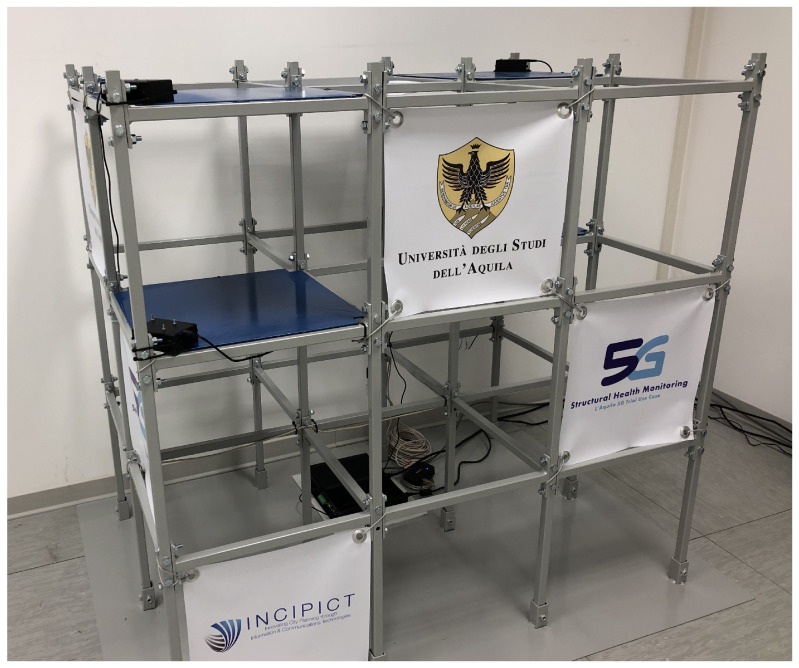
Test frame equipped with sensor boards and connected to a 5G CPE.

**Figure 6 sensors-22-03020-f006:**
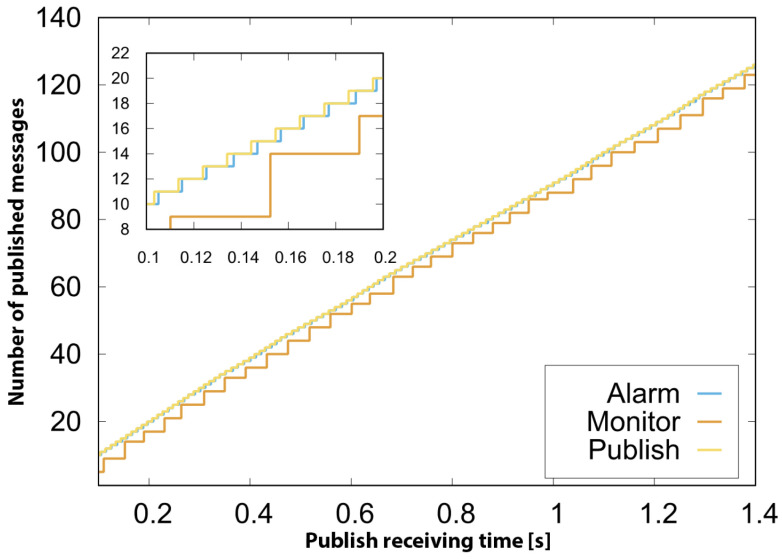
Publish messages receive time.

**Figure 7 sensors-22-03020-f007:**
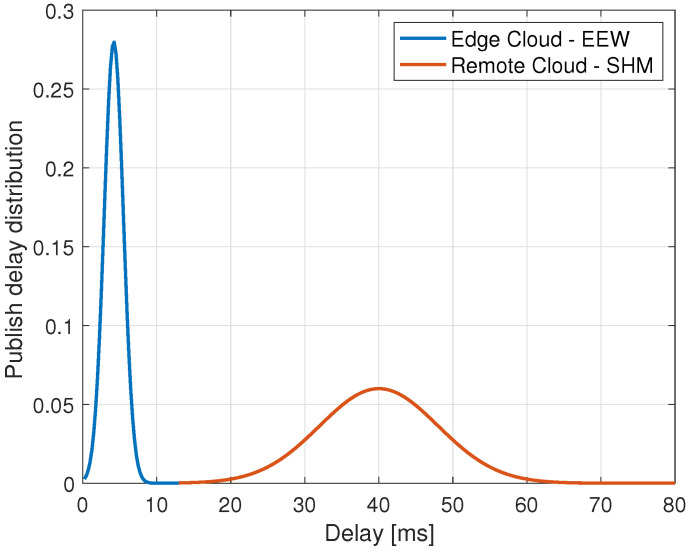
Distribution of delays at edge and remote cloud measured at the application layer.

**Figure 8 sensors-22-03020-f008:**
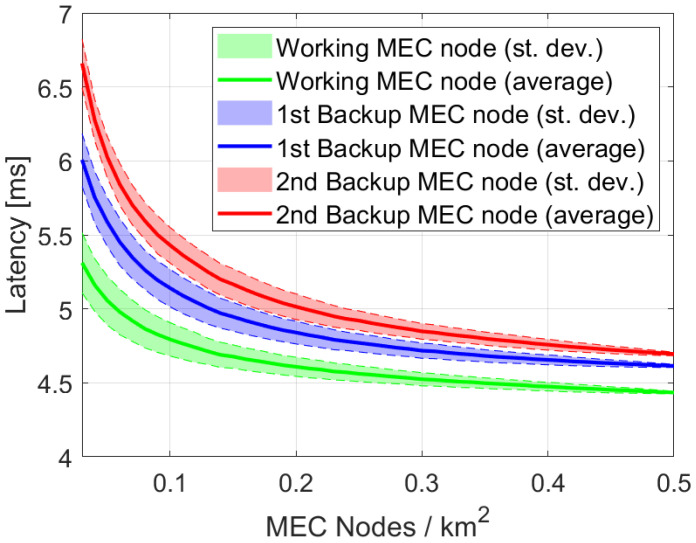
Average delays and standard deviations for working and backup MEC resources.

## Data Availability

Not applicable.

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
