# Peer review of "What Can 5G Do for Public Safety? Structural Health Monitoring and Earthquake Early Warning Scenarios"

_sensors, 2022, doi:10.3390/s22083020_

Round 1

Reviewer 1 Report

The authors studied the potential benefit of the 5G network to promote the operation of SHM systems and earthquake early warning (EEW) systems. The low latency and edge computing will enable the rapid EEW message faster than the earthquake and dedicated, reliable networks for SHM. The reviewer’s minor comments include:

  1. Lines 64 and 67. Errors in the citations. [?]
  2. Overall, the authors attempted to connect the 5G networks to SHM. It is hard to see the clear connections between the two components. Although software-defined, low-latency connections might help improve the EEW systems, it is unclear why the current SHM systems are failing due to the current 4G communications. What is the existing technological barrier for SHM in terms of wireless connections? What is the data size from the sensor nodes developed by the authors? The delay torrent networks and MQTT with the current 4G systems might be sufficient to support the SHM since most of the data analytics of SHM performs in a batch mode. Plus, structural conditions are not changing in a very short period of time (e.g., ms).
  3. Section 6 mostly validates the latency and delay of the 5G connections, which can be general. It is tough to see how the validation effort is related to SHM.
  4. What can be the significant improvement from 4G to 5G? It might be interesting if the authors could compare the test results between 4G and 5G connections.

Author Response

We thank the reviewer for all the valuable comments we received and we hope to have properly used them to improve the paper to a level of your satisfaction.
Please see the attachment for details.

Best wishes,
the authors

Reviewer 2 Report

  1. The abstract is too short and lacks a description related to research challenges/problems, methodology, 5G hardware settings, datasets measurement/generation, and impact of the study. The first 3 lines, i.e. Line 1 to 3 should be removed since the description was found to be very general and outdated related to 5G.  
  2. In Section 5, the sensor board design has been published in a previous conference [20], however, in this draft, the technical information provided should be given in full detail as well in the methodology section. Highlight how does the sensor implemented in the prototype is relevant to both SHM and EEW applications  
  3. In Section 1 of the manuscript, the authors should also describe how does this manuscript is different, or as an extension to the previous paper [20] 
  4. Line 64 to 67 - the references are missing and written as '?' instead 
  5. Unclear on the relationship between the developed sensor board prototype (Sect 5) and Section 7 (5G trial). Plenty of technical information is missing, including the followings:
    • Does the sensor board connect to the 5G network during the measurement?
    • How does the sensor board being deployed to represent actual SHM and EEW use cases? In other words, how the actual deployment of Figure 2 and Figure 4 has been done for this research? 
    • What kind of 5G network has been tested, whether 5G mid-band or lower band, the network architecture, distribution of the edge nodes etc.
    • Please provide relevant tables, drawings or/and discussions related to this technical 5G deployment. 
  6. How does the network latencies is measured in the study? Line 271-273
  7. The study focuses on URLLC via the deployment of edge cloud. In the result section, please make a comparison with other similar works interms of the delay performance.  
  8. Sect 8 (Conclusions) is too short. Please highlight the novelty/contributions and important findings from the study. There are no results or validations done to support the exploitation of the 6G in the scenario - pls remove this or rewrite it as future work by highlighting how it can be done.  

Author Response

(The authors gave the same response as above.)

Round 2

Reviewer 1 Report

The authors elaborated to improve the quality of the manuscript. They addressed this reviewer's previous concerns. 

Reviewer 2 Report

The authors have conducted revisions as suggested by the reviewer in the latest manuscript.